# Structure and mechanism of the RNA dependent RNase Cas13a from *Rhodobacter capsulatus*

Leonhard M. Kick[1], Marie-Kristin von Wrisberg[1], Leander S. Runtsch [ID][1] & Sabine Schneider [ID][1✉]

Cas13a are single-molecule effectors of the Class II, Type VI family of CRISPR-Cas systems that are part of the bacterial and archaeal defense systems. These RNA-guided and RNA-activated RNA endonucleases are characterized by their ability to cleave target RNAs complementary to the crRNA-spacer sequence, as well as bystander RNAs in a sequence-unspecific manner. Due to cleavage of cellular transcripts they induce dormancy in the host cell and thus protect the bacterial population by aborting the infectious cycle of RNA-phages. Here we report the structural and functional characterization of a Cas13a enzyme from the photo-auxotrophic purple bacteria *Rhodobacter capsulatus*. The X-ray crystal structure of the RcCas13a-crRNA complex reveals its distinct crRNA recognition mode as well as the enzyme in its contracted, pre-activation conformation. Using site-directed mutagenesis in combination with mass spectrometry, we identified key residues responsible for pre-crRNA processing by RcCas13a in its distinct catalytic site, and elucidated the acid-base mediated cleavage reaction mechanism. In addition, RcCas13a cleaves target-RNA as well as bystander-RNAs in *Escherichia coli* which requires its catalytic active HEPN (higher eukaryotes and prokaryotes nucleotide binding) domain nuclease activity. Our data provide further insights into the molecular mechanisms and function of this intriguing family of RNA-dependent RNA endonucleases that are already employed as efficient tools for RNA detection and regulation of gene expression.

[1] Ludwig-Maximilians University, Department of Chemistry, Institute for Chemical Epigenetics, Würmtalstr. 201, 81375 Munich, Germany.
✉email: sabine.schneider@cup.lmu.de

ype VI CRISPR-Cas systems are characterized by their large single-protein effector nuclease with a composite active site consisting of two highly divergent higher eukaryotes and prokaryotes nucleotide-binding (HEPN) domains. The type VI CRISPR-Cas systems most likely evolved from toxin-antitoxin systems through insertion of a HEPN-encoding element/gene next to a CRISPR array, duplication of HEPN domain and multiple insertions that increased specificity of crRNA binding as well as multiple independent acquisitions of the adaptation module[1]. These systems can be found in a number of bacterial species with Cas13a (formerly C2c2) as the hallmark enzyme[1], but share an overall low sequence identity of 20–30%[2]. All so far characterized Cas13-family members possess two distinct active sites, one for pre-crRNA processing and one for target RNA cleavage[3–6]. They become activated by sequence-specific binding of a single-stranded target RNA to the crRNA, which induces a conformational change and alignment of the catalytic tetrad, in the composite HEPN domains, resulting in, cleavage of the target ssRNA (cis-cleavage) and sequence-unspecific bystander ssRNAs (trans-cleavage)[3,4,7–10], which has been exploited for the in situ detection of viral RNAs[11]. Additionally in *Listeria* it was shown that activation of Cas13a leads to trans-cleavage of mRNA transcripts, that in turn leads to induction of dormancy in the host, which is sufficient to abort lytic infection and phage propagation in a cellular population[12]. This resembles the strategy of abortive infection systems where dormant enzymes are activated upon phage replication, resulting in inactivation of the cellular translation or transcription machinery, protein phosphorylation or alteration of the membrane integrity, leading to growth arrest or cell death, thus arresting phage infection and spreading[13]. Moreover, phages with target mutations that evade DNA-targeting by CRISPR-Cas systems are neutralized by Cas13a when activated by wild type phages[14–17]. Therefore, CRISPR-Cas13a systems prevent outbreaks of CRISPR-resistant phages by acting on the host and not directly on virus[12].

Here we provide data on the structure and mechanisms of the Cas13a-endonuclease from the photo-auxotrophic bacteria *Rhodobacter capsulatus* (RcCas13a), which shares ~20–25% sequence identity with the to date structurally characterized homologous enzymes from *Lachnospiraceae bacterium* (Lba)[10], *Leptotrichia buccalis* (Lbu)[9], *L. shaii* (Lsh)[9], and *Listeria seeligeri* (Lse)[12]. The X-ray crystal structure of RcCas13a in complex with its crRNA elucidates how RcCas13a specifically recognizes its crRNA and shows the enzyme in a contracted, pre-activation conformation. In order to determine amino acid residues responsible for pre-crRNA processing by RcCas13a in the not conserved likely processing catalytic site, we employed site-directed mutagenesis in combination with in vitro activity assays and mass spectrometry. This allowed us not only to identify key residues responsible for pre-crRNA processing by RcCas13a, but also to elucidate the reaction mechanism. Moreover, we could show that RcCas13a can be activated in *Escherichia coli*, where it cleaves target RNA as well as bystander-RNAs, which requires a catalytic active domain. In summary, this study provides further insight into the molecular mechanisms of this intriguing family of RNA-dependent RNA endonucleases.

## Results and discussion
### Overall structure of the RcCas13a-crRNA binary complex.
The reference genome of *R. capsulatus* contains three CRISPR-Cas systems: Two Class I systems (Type I and Type III) as well as one Class II, Type VI system (Supplementary Fig. S1), with the Cas13a-signature sequence encoding a 145 kDa protein sharing the family-typical HEPN-endonuclease domains, with the

**Table 1 Data collection, processing, structure solution, and refinement statistics.**

|  | Native | SeMet |
|---|---|---|
| Data collection |  |  |
| Space group | P1 | P1 |
| Cell dimensions |  |  |
| *a*, *b*, *c* (Å) | 61.4, 91.1, 136.5 | 62.6, 91.2, 137.1 |
| α, β, γ (°) | 90.0, 103.7, 97.6 | 90.0, 103.5, 96.7 |
| Resolution (Å) | 47.1–2.2 (2.3–2.2)[a] | 47.7–2.5 (2.7–2.5) |
| $R_{merge}$ | 0.085 (0.913) | 0.153 (0.742) |
| $I/\sigma I$ | 9.6 (1.2) | 8.9 (1.6) |
| Completeness (%) | 97.6 (96.3) | 96.5 (89.1) |
| Redundancy | 3.6 (3.4) | 6.3 (3.9) |
| $CC_{1/2}$ | 0.997 (0.726) | 0.995 (0.745) |
| $CC^a$ | 0.999 (0.917) | 0.999 (0.924) |
| Refinement |  |  |
| Resolution (Å) | 47.1–2.2 |  |
| No. reflections | 504,148 (48,268) |  |
| $R_{work}/R_{free}$ | 0.218/0.258 |  |
| No. atoms |  |  |
| Protein | 17,795 |  |
| Ligand/ion | 1986 |  |
| Water | 630 |  |
| *B*-factors |  |  |
| Protein | 94.2 |  |
| Ligand/ion | 79.9 |  |
| Water | 51.6 |  |
| R.m.s. deviations |  |  |
| Bond lengths (Å) | 0.005 |  |
| Bond angles (°) | 0.88 |  |

[a]Values in parentheses are for highest-resolution shell. Data were collected on one single crystal each.

conserved Arg-His-Arg-His catalytic tetrad (Fig. 1a and Supplementary Fig. S2)[18]. We expressed RcCas13a alone and together with a CRISPR-locus in *E. coli* and purified apo-RcCas13a as well as the enzyme in complex with the ~58 nt crRNA (RcCas13a-crRNA binary complex). The RcCas13a-crRNA binary complex exhibits a markedly increased stability of about 10 °C (Supplementary Figs. S3 and S4). We determined the X-ray crystal structure of the binary complex by experimental phasing, using crystals of SeMet-labeled RcCas13a protein and refined the final coordinates against data collected on native RcCas13a-crRNA complex crystals to 2.1 Å resolution (for X-ray data processing, structure solution and refinement statistics see Table 1). Protein residues as well as nucleobases of the repeat-region and the first 16nt's of the spacer in the crRNA are well defined in the electron density. However, some nucleobases in the middle of the spacer sequence and solvent-exposed loop regions in the protein are flexible and partly disordered (Supplementary Fig. S5). RcCas13a shares the common domain architecture and predominantly α-helical-folding topology of the Cas13a-family. (Fig. 1a, b and Supplementary Fig. S6) However, we did not observe electron density for the C-terminal 120 aa residues, that are unique to RcCas13a (Supplementary Fig. S2). Using HHPRED[19] we recognized that the C-terminal domain (CTD) has similarity to members of the cold shock domain-protein family, which are multifunctional RNA/DNA binding proteins, exhibiting a β-barrel fold[20] (Supplementary Fig. S7). Albeit the sequence identity is low (~15%), it is likely that the RcCas13a-CTD adopts a similar overall fold. Nevertheless, all our attempts to express and purify solely the RcCas13a-CTD domain for structural characterization failed.

The binary complex structure of RcCas13a shows the enzyme in a contracted, inactive conformation. Here, in the composite

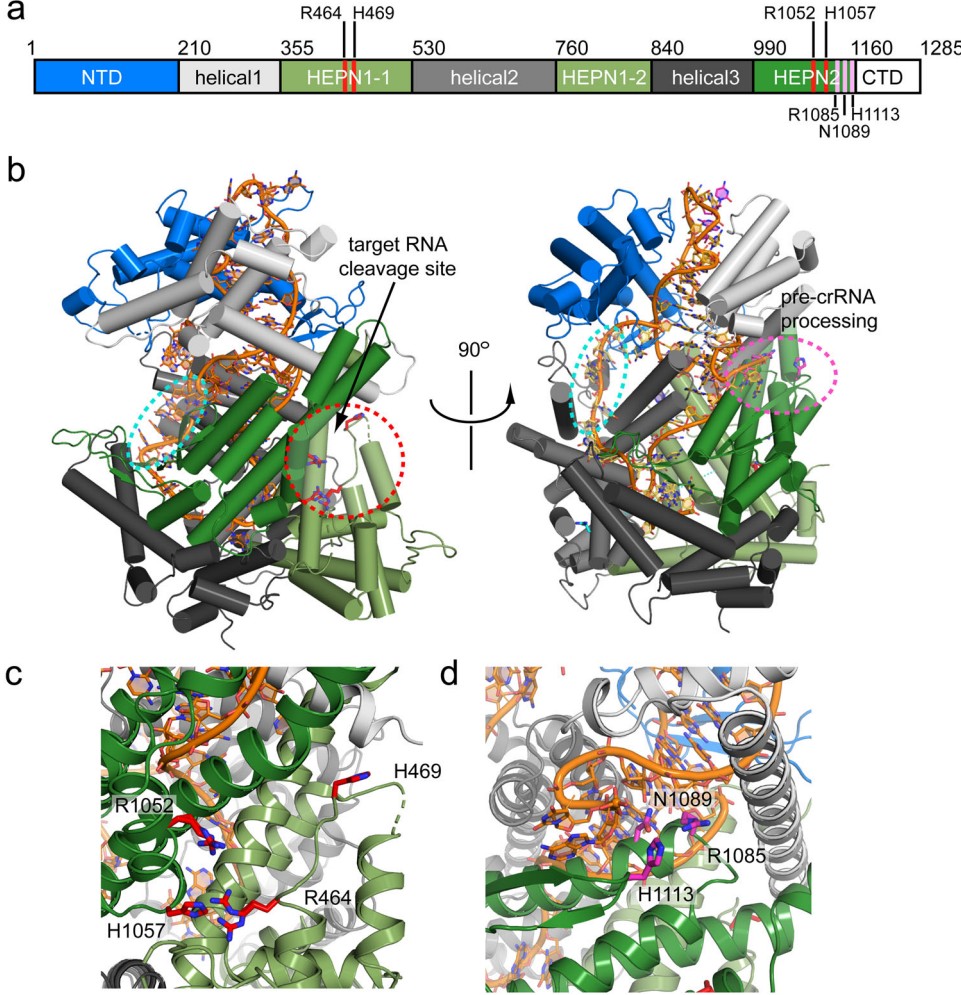

**Fig. 1 Overall structure of the RcCas13a-crRNA complex. a** Domain structure of RcCas13a with the residues important for the catalytic activity highlighted in red (trans-ssRNA cleavage) and pink (pre-crRNA processing), respectively. **b** The location of the two independent active sites are indicated by a red circle, the seed region of the crRNA (position 10–14) by a cyan circle. **c** Zoom in the RNase active site, with the residues crucial for catalytic activity (Arg 464/His 469 and Arg 1052/His 1057) highlighted as red stick model. **d** Close-up view of the pre-crRNA processing site, with Arg 1085, Asn 1089 and His 1113 essential for pre-crRNA cleavage (see below) highlighted in pink. The crRNA is shown as orange stick model.

active site made up by the HEPN1 and HEPN2 domains, His 469 of the conserved catalytic tetrad (Arg 464 + His 469/Arg 1052 + His 1057) is shifted about 18 Å away, preventing endonuclease activity prior to target RNA binding (Fig. 1b, c). Albeit HEPN-domains do not share a significant sequence conservation, they are characterized by an α-helical-folding topology and a conserved Arg and His sequence motif $(R(x)_{4-6}H$ (where x can be any residue; Supplementary Fig. 8). Two HEPN-domains have to align the Arg and His residues, either through dimerization of the two domains contained within one protein, like in Cas13 endonucleases, or of two HEPN-domain proteins, two form the catalytic tetrade required for metal-independent endonuclease activity[21,22]. As common for type VI endonucleases, RcCas13a possesses two distinct active sites for crRNA processing and target RNA cleavage (Fig. 1c, d). This is in agreement with functional and structural studies on the homologs with ~23% sequence identity to RcCas13a which demonstrated that target RNA binding to the binary complex induces a conformational change and rotation of the helical 2 (Hel 2) domain relative to the HEPN1 domain, resulting in rearrangement of the catalytic tetrad thereby switching-on the RNase activity (Supplementary Fig. S9)[9,10,12]. Thus, the stable contracted conformation of the binary complex as observed in

RcCas13a, likely prevents premature activation of the RNase function.

**Recognition of the crRNA by RcCas13a.** The overall common crRNA structure of the binary Cas13a-crRNA complexes consists of a stem-loop, a 2 nt bulge at the 3′ stem and an A-U base pair at the bottom of the stem, implying a shared shape readout mechanism of crRNA recognition by this endonuclease family, combined with some sequence-specific base-readout. RcCas13a recognizes its ~58 nt long crRNA predominantly via shape-recognition, but makes nucleobase-specific interactions with its stem-loop and 3′ flank of the crRNA. The peptide backbone in the helical I (Hel 1) domain (resi 276–279) contacts the exocyclic carbonyl group in C-17 and the Hoogsteen-face of G-16 in the loop (Fig. 2c). The ε-amino group of Lys 179 interacts with [7]N of A-15, and Asp 293 contacts the exocyclic amino group in A-21 (Fig. 2a, b). In addition, a base-specific readout of the bulge formed by A-8 and C-7 at the 3′ base of the stem-loop occurs via bi-dental hydrogen-bonding of Arg 23 and the peptide backbone of the β-hairpin in the NTD to the Watson–Crick–Franklin face of C-7 (Fig. 2d). The exocyclic amine in A-8 is interacting with the carbonyl oxygen of Ser 684 located in α-helix of the Hel 2 domain. Moreover, G-5 and C-1 in the 3′ flank of the crRNA

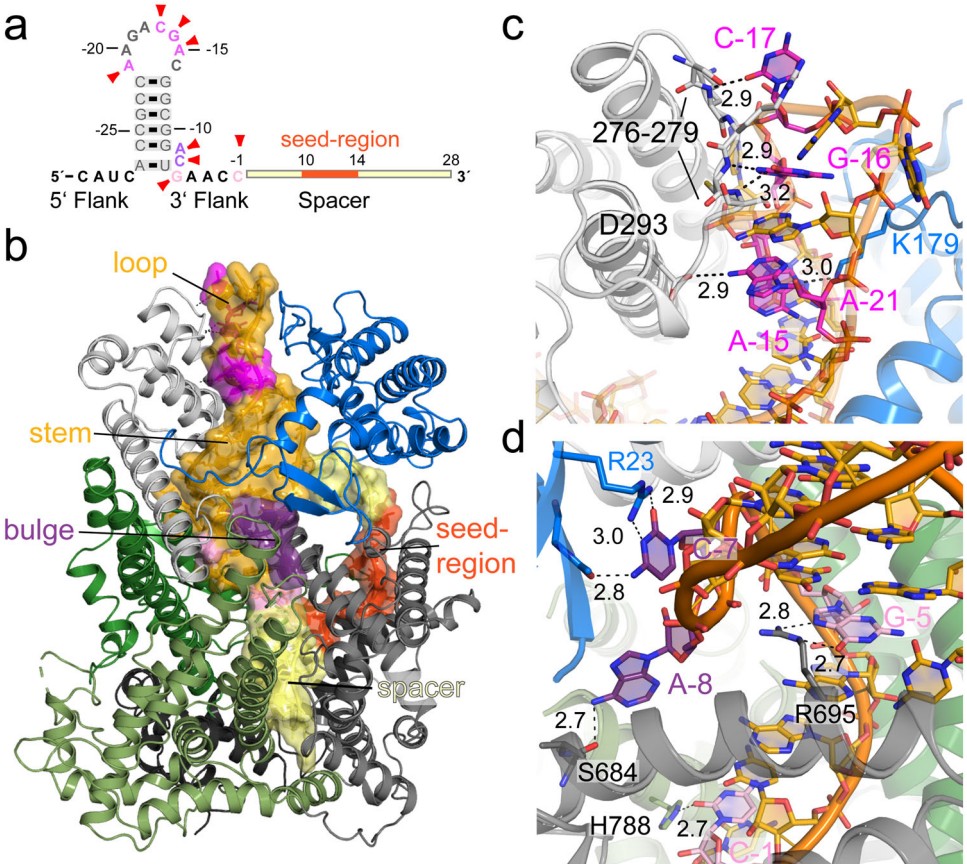

**Fig. 2 RcCas13a-crRNA interactions. a** Schematic depiction of the crRNA, with base-specific contacts indicated by red arrows. **b** Overall RcCas13a complex with the crRNA overlaid with its semi-transparent surface. RcCas13a is colored according to its domain as in Fig. 1. cRNA: stem-loop = golden; spacer = light yellow; seed region = orange; sites with base-specific interaction are highlighted in pink and violet. **c** RcCas13a base-specific interaction with the tip of the crRNA hairpin, **d** with the crRNA bulge at the base of stem and C-1 in the 3′ flank.

form interactions with Arg 695 (Hel 2) and His 788 (HEPN1-2) (Fig. 2d). All other observed interactions of RcCas13a and its crRNA occur in a nucleobase-unspecific manner and the phosphodiester backbone of the crRNA (Supplementary Figs. S10 and S11). Distinct to the crRNAs from Lba, Lbu, Lse and LshCas13a, the stem-loop in the crRNA of RcCas13a is extended by one C-G base pair and an additional 3.4 Å interaction between the twisted C-28 ($^3$N) and G-5 ($^1$N) at the base of the stem, next to the conserved U-A base-pair can be observed (Supplementary Fig. S11). Moreover, RcCas13a does not form any sequence-specific contacts with the 5′ flank of the crRNA. In contrast, LbuCas13a and LseCas13a interact base-specific with an A and a G, respectively, in the 5′ flank. It was shown for LbuCas13a that it is sensitive to mutation in the 5′ flank of the crRNA as well as to mutations that impact on the overall shape of the stem-loop[3,12]. Nucleobases at positions 10–14 in the crRNA-spacer region are solvent exposed and are presumably acting as the seed region for target RNA binding (Fig. 1b), as shown for the homolog LbuCas13a[23]. The 2-nt bulge at the base of the 3′ stem-loop, a common characteristic of so far investigated Cas13a endonucleases (Supplementary Fig. S11), was shown to be important to regulate LseCas13a and LshCas13a activity. Here the extended sequence complementarity and base pairing between the crRNA and target RNA by 8 nucleotides (= anti-tag) blocks activation of the endonuclease. This so called tag:anti-tag interaction appears to allow discrimination between self and non-self, preventing erroneous activation of the endonuclease[24,25]. A consequence of the extended A-form dsRNA formed between crRNA and target RNA parts of the hairpin and the bulge of the crRNA

disintegrate, leading to a misalignment of HEPN 1 + 2-domain catalytic residues and inactive enzyme[25].

**Mechanism of pre-crRNA processing**. For some Cas endonucleases from the Cas13a and Cas12a/Cpf1-families, it was shown that they process their own pre-crRNA in a distinct active site[9,10,26], while Cas9-homologs rely on RNaseIII, a general RNA processing factor in bacteria, for crRNA biogenesis[27]. We investigated the crRNA processing activity of RcCas13 in vitro, using synthetic 3′ 6-Carboxyfluorescein (6-FAM) labeled pre-crRNA. Our experiments show that RcCas13a does not rely on the catalytic HEPN-domains for pre-crRNA processing since RcCas13a mutants, where the active-site residues in one or both of the HEPN1 or HEPN2 domains were mutated to Ala were not affected in their pre-crRNA processing activity (Fig. 3a). Moreover, once bound to RcCas13, the crRNA is not released from the protein again since the binary complex does not cleave 6-FAM-labeled pre-crRNA (Fig. 3a). Furthermore, pre-crRNA processing by RcCas13a is independent of metal ions and the enzyme is still active in the presence of the chelator EDTA in the reaction buffer (Fig. 3b). This indicates an acid-base-catalyzed processing mechanism, in agreement with previous studies on some Cas13a homologs[9,10] and Cas12a/Cpf1[26]. In order to gain deeper insights into the pre-crRNA processing mechanism of RcCas13a, we mutated all likely candidate amino acid residues located in the vicinity of the 5′ end of the crRNA in the binary complex structure, since sequence conservation in this area is limited. Following expression and purification, the integrity of all purified

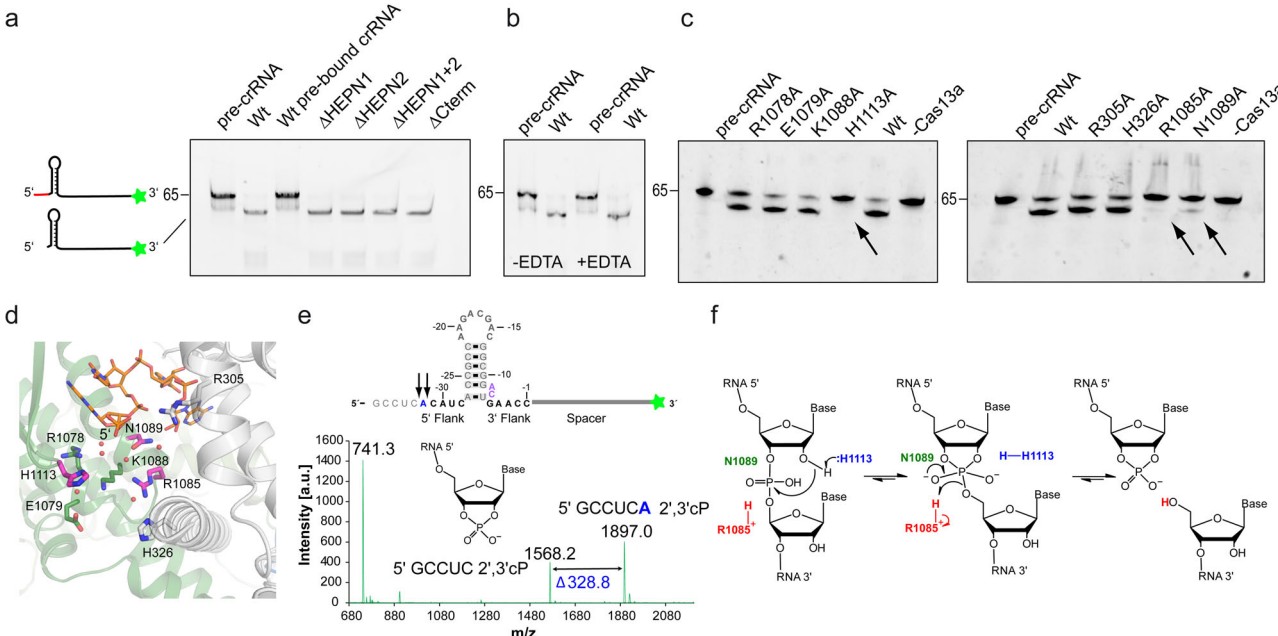

**Fig. 3 Processing of pre-crRNA by RcCas13a. a** RcCas13a cleaves 3′-FAM-labeled pre-crRNA (65 mer) independent of its catalytic HEPN-domains or the C-terminal domain. RcCas13a pre-bound to crRNA does not process 3′-FAM-labeled pre-crRNA. ΔCTD = Δ C-terminal domain; H1* = RcCas13a$^{R464A/H469A}$; H2* = RcCas13a$^{R464A/H469A}$; H1 + 2* = RcCas13a$^{R464A/H469A/R1052A/H1057A}$. **b** Pre-crRNA cleavage is not inhibited by EDTA and hence metal-ion-independent. **c** Identification of residues involved in pre-crRNA processing by mutational analysis. **d** Close-up view of the 5′ end of the crRNA in the RcCas13a binary structure. Residues in the vicinity of the 5′ end of the crRNA and subject to mutational analysis are shown as stick model, with the three residues crucial for pre-crRNA processing highlighted in pink. Water molecules are shown as spheres. **e** MALDI-TOF analysis of the pre-crRNA processing reaction shows cleavage of the pre-crRNA 4 or 5 nt 5′ of the stem-loop, resulting in a 2′,3′-cyclic 3′ phosphate (2′,3′-cP). **f** Proposed mechanism of pre-crRNA processing by acid-base catalysis. His 1113 acts as the base, abstracting the proton of the 2′OH next to the scissile phosphate. Arg 1085 acts as a general acid and protonates the 5′-oxygen leaving group.

proteins was analyzed by dynamic light scattering (DLS) to exclude that the introduced mutations have an impact on protein stability and hence catalytic activity, rather than the chemical properties of the respective amino acid side chains (Supplementary Fig. S12).

We tested all RcCas13a mutants for their pre-crRNA cleavage activity and could identify Arg 1085, Asn 1089 and His 1113 as key residues essential for pre-crRNA processing. Mutation of these residues to Ala completely or, in case of Asn 1089 partially, abrogates pre-crRNA processing activity (Fig. 3c). While Arg 1085 and Asn 1089 are located at the N-terminus of an α-helix in HEPN2 at the bottom of a cleft between the Hel 1 and HEPN2-domain, His 1113 sits at the tip of a β-hairpin in the HEPN2-domain (Fig. 3d). While in LbuCas13a Arg 1079, Lys 1082 and Asn 1083 residues at structurally equivalent positions at the N-terminus of the α-helix where shown to be crucial in LbuCas13a for pre-crRNA cleavage (Supplementary Fig. S13)[4,9], mutation of the corresponding Lys 1088 in RcCas13a did not influence its pre-crRNA processing activity (Fig. 3c and Supplementary Fig. S15). While in the homolog from *L. shahii* (LshCas13a) Arg 438 and Lys 441 located in the NTD are responsible for pre-crRNA cleavage[9], Dounda co-worker report for LbaCas13a that Lys 1305 and Asn 1232 located in the HEPN2 domain are required for pre-crRNA processing[10]. Additionally, in the LbaCas13a complex the 5′ flank of the pre-crRNA and crRNA are tightly held by the surrounding amino acid side chains. In contrast, the equivalent site in the RcCas13a complex is more open, allowing more flexibility for the 5′ flank of the crRNA (Supplementary Fig. S14). Nevertheless, LbaCas13a and RcaCas13a both utilize residues located in the HEPN2 domain for pre-crRNA processing. Cas13d from uncultured *Ruminococcus sp.* (UrCas13d), another member of the Type VI family of Cas

endonucleases, also requires two residues, Arg 802 and Lys 905 located in its HEPN2 domain for pre-crRNA processing[5]. These are to our knowledge to date the only examples of catalytic activity HEPN domains that contain a second RNase active site (Supplementary Fig. S8)

Although all Cas13a homologs whose structures have been determined share the same overall domain structure, they vary in the relative orientation of the domains as well as the exact positioning of the crRNA (Supplementary Fig. S6). In general, metal-free acid-base catalysis of nucleases are $S_N2$ reactions. Here a nucleophile attacks the scissile phosphate either from the 3′ side, breaking the 5′ O-P bond, which produces 3′-phosphate and 5′-OH, or the 5′ side, breaking the 3′ O-P bond, resulting in a 5′-phosphate and 3′-OH[28]. To gain more insights into pre-crRNA processing by RcCas13a, we analyzed the reaction products of the in vitro pre-crRNA processing reaction by matrix-assisted laser desorption ionisation-time of flight (MALDI-TOF) mass spectrometry, which possesses much higher sensitivity and precision in comparison to gel-electrophoresis. This revealed that RcCas13a cleaves either 4 or 5 nt 5′ to the stem-loop, generating a 2′,3′-cyclic 3′ phosphate (2′,3′ cP, Fig. 3e and Supplementary Fig. S15). Given that the reaction product is a 2′,3′-cP, the 2′-oxygen next to the scissile phosphate has to carry out the nucleophilic attack and is thus transferred to the 5′ phosphate. Substitution of the ribose by the 2′ deoxyribose (dA or dC) in synthetic pre-crRNA strands abrogates cleavage at this positions (Supplementary Fig. S15), providing further support for the importance of the nucleophilic attack by the 2′OH. Prerequisite for this transfer reaction is deprotonation of the 2′OH group, possibly by the essential His 1113 acting as the base. However, it is also possible that His 1113 abstracts a proton from a water molecule and that the resulting hydroxide ion acts as the effective base, since our structure shows

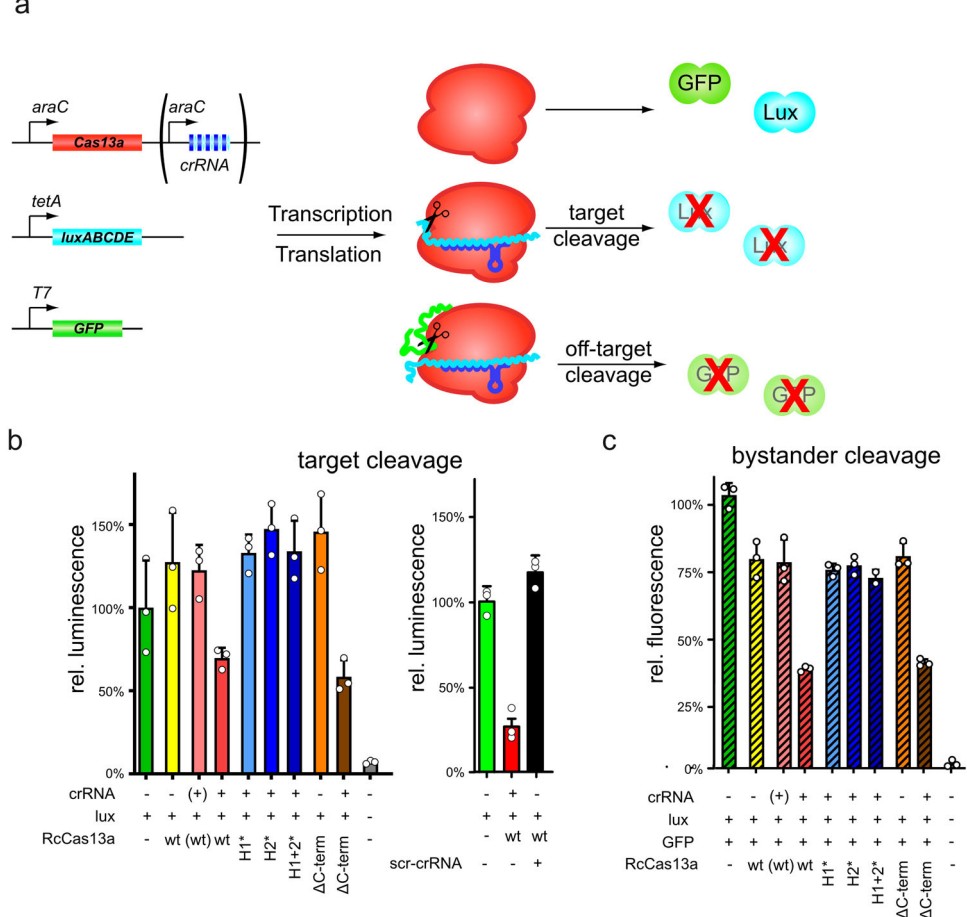

**Fig. 4 In vivo activity of RcCas13a. a** Schematic depiction of the in vivo activity assay. Two reporter gene constructs (*luxABCDE* and *GFP*) as well as expression constructs for RcCas13a variants and crRNA encoded within a CRISPR-locus targeting *luxABCDE* are introduced into *E. coli* cells. Upon expression of RcCas13a and crRNA*luxA* cis-cleavage activity can be monitored by reduction of luminescence, while sequence unspecific *trans*-cleavage can be assessed by alteration in GFP-fluorescence. **b** Sequence-specific target cleavage activity (*cis*-cleavage) by RcCas13a is followed by bioluminescence upon luciferase expression reduction. Wt RcCas13a and RcCas13aΔCTD reduce bioluminescence in a crRNA*luxA* dependent manner (red and brown bar), while all HEPN-mutants lost their impact on luciferase expression (blue bars). Scrambling of the sequence in the seed region nt 10–14 of the spacers abrogates reduction of luciferase level by RcCas13a (black bar). **c** Sequence-unspecific RNA cleavage (bystander-cleavage) was analyzed by measuring GFP-fluorescence upon induction of crRNA*luxA* and target RNA expression (*luxABCDE*). Wt RcCas13a and RcCas13aΔCTD reduce GFP-fluorescence only upon expression of crRNA*luxA* and target sequence (red and brown hatched bars). Mutation of any of the catalytic residues in the HEPN-domains leads to an inactive RcCas13a (blue hatched bars). RcCas13a and crRNA expression is induced by addition of arabinose to the growth medium. Luciferase and GFP expression is controlled by anhydrotetracycline and IPTG, respectively. (+): Encoding gene/expression construct is present in the *E. coli* cells, but was not induced. (−) empty vector control. (H1* = RcCas13a$^{R464A/H469A}$; H2* = RcCas13a$^{R464A/H469A}$; H1 + 2* = RcCas13a$^{R464A/H469A/R1052A/H1057A}$; ΔCTD = Δ C-terminal domain); experiments were performed in three independent replicates (*n* = 3).

RcCas13a in complex with the processed, mature crRNA and not the pre-crRNA bound state. Asn 1089 could play a role in the stabilization of the penta-covalent reaction intermediate and/or positioning of the 5′ flank, since mutation of Asn 1089 to Asp does not impact on the cleavage position but mutation to Ala leads to aberrant cleavage 6 nt's 5′ to the stem-loop (Supplementary Fig. S15). Arg 1085 could act as the acid and protonates the 5′-oxygen leaving group (Fig. 3f). Due to the high intrinsic pKa value of the Arginine's guanidinium group (pKa 13.8 ± 0.1) and delocalization of the positive charge over many atoms involved in a conjugated π-system, the arginine side chain has the unusual ability to retain its charge under all physiological conditions and can thus function as an acid catalyst[29,30]. As already observed by gel-electrophoresis, the RcCas13a$^{R1085A}$ mutant does not show any pre-crRNA cleavage activity in the MALDI-analysis of the reaction products (Supplementary Fig. S15). As stated above, in our mutational analysis of residues lining the pre-crRNA processing site in RcCas13a, we could

identify Arg 1085 and His 1113 as being clearly essential, and Asn 1089 modulating crRNA cleavage. In contrast, for LbaCas13a, which possesses a narrower pre-crRNA binding site (Supplementary Fig. S15), it was shown that Trp 325 and Asn 1232 are the likely candidates to set up the 2′OH of the ribose of the scissile base in the pre-crRNA for nucleophilic attack. Than an activated water molecule, together with Lys 1305, His 328, Lys432, and Lys 1320 participate in the proton transfer reactions[10].

**In vivo activity of RcCas13a.** We investigated the in vivo cleavage activity of RcCas13a in *E. coli* JM109 using plasmid encoded reporter genes (luciferase: *luxABCDE*; green fluorescence protein: *GFP*) under control of an anhydrotetracycline P$_{TET}$ and an IPTG inducible T7-promoter, respectively. RcCas13a and a crRNA with complementary spacer sequence to *luxA* are under transcriptional regulation of the arabinose responsive P$_{BAD}$-promoter (Fig. 4a). This setup allows the distinct expression of all assay components:

RcCas13a (wt or mutants), crRNA$^{luxA}$, luciferase (reporter for *cis*-cleavage activity), gfp (reporter for *trans*- or bystander cleavage activity). Upon expression of wt RcCas13a and crRNA we observe a reduction in bioluminescence due to the sequence-specific degradation of *luxABCDE* mRNA by RcCas13a (Fig. 4b, left red bar, *cis*-cleavage), which is independent of expression induction of the second reporter *gfp* (Supplementary Fig. S16). This observed reduction in bioluminescence is strictly dependent on the presence and expression of the *luxA*-specific crRNA (Fig. 4b, yellow and pink bars). Deletion of the CTD in Cas13a does not affect RcCas13a activity (Fig. 4b, brown bar), while mutation of any of the catalytic residues in the HEPN-domains abolishes RcCas13a activity (Fig. 4b, blue bars). Sequence independent *trans*-cleavage was analyzed by GFP-fluorescence. Wt RcCas13a and RcCas13aΔCTD reduce GFP-fluorescence only when co-expressed with crRNA and *luxABCDE* (Fig. 4c, red and brown hatched bars). Without co-expression of either crRNA or the *lux*-operon GFP- fluorescence remained unaffected by RcCas13a expression (Fig. 4c, yellow hatched bar and Supplementary Fig. S16). Again RcCas13a-mediated reduction in GFP-fluorescence is dependent on the HEPN-domains and prevented by mutation of any of the active site residues (Fig. 4c, blue hatched bars). This HEPN-dependent *cis*- and *trans*-cleavage activity by RcCas13a is common to the Cas13a-family[3,7,9,10,12]. Scrambling of the sequence 10–14 of the spacer (scr-crRNA), which was shown for other Cas13a homologs[24] acting as seed region for interaction between crRNA and target RNA, abolishes reduction of luciferase luminescence upon expression of RcCas13a scr-crRNA (Fig. 4b, right, black bar). Moreover, expression of wt RcCas13a (or RcCas13aΔCTD) in combination with crRNA and target RNA leads to a reduced bacterial growth of the *E. coli* cells (Supplementary Fig. S17). This is in agreement with previous studies on Type VI CRISPR-Cas systems, showing that Cas13a mediates protection against phages by dual function: first, by direct target RNA degradation and secondly, abortion of the infectious cycle by inducing growth arrest (dormancy) in the host cell through *trans*-cleavage of transcripts. With this indirect targeting mechanism Type VI systems also act on CRISPR-resistant phages, deplete the phage population and thus provide herd immunity to uninfected bacteria[25,31].

## Conclusion

The photo-auxotrophic bacterium *R. capsulatus* encodes, next to Class 1, also a Class 2 Type VI CRISPR-Cas system, harboring a 145 kDa Cas13a single-effector endonuclease. This RcCas13a possess two distinct active sites for crRNA processing and ssRNA endonuclease activity that are typical for this family. The X-ray crystal structure of RcCas13a in complex with its crRNA reveals, that the enzyme combines shape and sequence-specific readout to recognize its crRNA and shows the enzyme in its stable and contracted, inactive conformation, which likely prevents premature activation of the endonuclease activity. Moreover, RcCas13a processes its crRNA, by cleaving either 4 or 5 nt upstream of the stem-loop, which results in a 2′,3′ cyclic phosphate at the 5′ leaving group. Crucial for the crRNA processing activity are three residues located in the processing site: His 1113 and Arg 1085 likely acting as acid and base. Mutation of Asn 1089 to Ala influences the position of the pre-crRNA and resulting in a partial shift in cleavage site preference. Upon expression in *E. coli*, RcCas13a utilizes its HEPN-domain composite active site to cleave target RNAs with complementary sequence to the crRNA-spacer sequence. The presence of crRNA and complementary target RNA also unlocks the HEPN-domain dependent *trans*-cleavage activity of bystander RNAs and results in a reduced bacterial growth, indicating that RcCas13a protects the bacterial population by aborting lytic infection and phage propagation by inducing dormancy in the host, as reported for homologs. In summary, our data provide further insights into the structure, mechanism and function of this intriguing family of RNA-dependent endonucleases.

## Methods

**Molecular biology.** Protein and nucleic acid sequences (RcCas13a, RNA, plasmids and primers) are listed in Supplementary Tables S1–S4. Mutants were prepared using standard cloning procedures (restriction enzymes (New England Biolabs), Gibson Assembly, NEBuilder HiFi DNA Assembly (NEB), QuikChange® Mutagenesis (Agilent), Q5 site-directed mutagenesis (NEB), SLIM PCR[32]) according to the protocol of manufacturer's instructions. All used plasmids were verified by Sanger sequencing (GATC Biotech/Genewiz, Germany).

**Protein expression of RcCas13a and RcCas13a mutants.** For protein expression RcCas13, annotated as a hypothetical protein (gene bank accession code WP_013067728.1), or RcCas13a mutants were cloned into the respect expression plasmid (see Supplementary Table S3) and expressed in *E. coli* Rosetta2 (DE3). Cells obtained from an overnight culture (LB medium supplemented with respective antibiotics according to Supplementary Table S3, 37 °C, 180 rpm) were diluted 1:100 (OD$_{600}$ between 0.05–0.07) in 2YT medium (50 µg/ml kanamycin) and incubated at 37 °C, 180 rpm until an OD$_{600}$ of 0.7 was reached. After shock-cooling for 15–30 min on ice, protein expression was induced with 1 mM isopropyl β-D-1-thiogalactopyranoside (IPTG) and incubated overnight at 18–20 °C, 180 rpm. Cells were subsequently harvested by centrifugation (5000 × *g*; 10 min; 4 °C).

**RcCas13a and RcCas13a-crRNA complex purification.** All purification steps were performed at 4–8 °C or on ice unless otherwise noted. The two-step purification procedure of RcCas13a proteins comprised a Ni$^{2+}$-affinity chromatography followed by removal of the His$_6$-MBP-tag using *Tobacco Etch Virus* (TEV) protease and a size-exclusion chromatography (SEC), both carried out on an ÄKTA-FPLC system (GE Healthcare, now Cytiva). The cell pellet obtained from a 1-liter expression was resuspended in 30 ml His-wash buffer (50 mM 4-(2-hydroxyethyl)-1-piperazineethanesulfonic acid (Hepes) pH 8.0, 1 M NaCl, 10% (v/v) glycerol, 1 mM dithiothreitol (DTT), 10 mM imidazole) supplemented with 0.1 mg/ml DNase I (AppliChem), 0.1 mg/ml lysozyme (~20,000 U/mg; Carl Roth) and one tablet cOmplete$^{TM}$ ULTRA EDTA-free protease inhibitor tablet (Roche). Cells were lysed by homogenization using an EmulsiFlexC5 (Avestin Inc.) or by sonication and cleared lysate was obtained by centrifugation (24,446 × *g*; 30 min; 4 °C) and filtration using a Whatman$^{TM}$ folded filter (GE Healthcare, now Cytiva).

Affinity purification used a 1 ml-HisTrap FF crude column (GE Healthcare, now Cytiva) at a flow rate of 1 ml/min. After sample application the column was washed with 12 CV His-wash buffer followed by 5 CV of His-wash buffer supplemented with imidazole to a final concentration of 30 mM. RcCas13a proteins were eluted with 13 CV His-elution buffer (50 mM Hepes pH 8.0, 500 mM NaCl, 10% (v/v) glycerol, 1 mM DTT, 300 mM imidazole) and collected in 0.5 ml fractions. Fractions containing the protein were pooled and concentrated with an Amicon Ultracell Centrifugal filter unit (MWCO 50 kDa, Merck Millipore) to a volume of 500 µl and digested with TEV protease for 2 h at 8 °C. After centrifugation (16,000 × *g*; 10 min; 4 °C) the sample was applied onto a HiLoad Superdex 200 (16/60) column (GE Healthcare, now Cytiva). Separation was performed at a flow rate of 0.8 ml/min in SEC buffer (50 mM Hepes pH 8.0, 500 mM NaCl, 10% (v/v) glycerol, 1 mM DTT) and 1.0 ml fractions were collected. Fractions containing the respective monomeric RcCas13a proteins were pooled, concentrated, and either directly used for crystallization trials or flash-frozen in liquid nitrogen and stored at −80 °C. The RcCas13-crRNA binary complex was obtained by co-expressing RcCas13a containing a C-terminal His$_6$-tag with the CRISPR-locus encoded on the same plasmid (pET24-RcCas13a+crRNA). The co-purification of the RcCas13a-crRNA complex was done analogously to the RcCas13a protein, without removal of the His-tag.

**Expression and purification of the SeMet-labeled RcCas13a-crRNA binary complex.** For production of SeMet-labeled protein, was carried out according to the modified protocol by van den Ent and Löwe[33]. In short, a culture of *E. coli* Rosetta2 (DE3) was grown in M9 minimal medium at 37 °C under agitation until an OD$_{600}$ of 0.6 was reached, where an amino acids mix (final concentrations: 100 mg/l-lysine, threonine and phenylalanine, 50 mg/l-leucine, isoleucine, valine and L-selenomethionine) was added as solids and incubated for 15 min. Cells were shock-cooled for 30 min at 4 °C and protein expression was induced with 1 mM IPTG overnight. Purification was performed as for the unlabeled protein, but using 5 mM β-mercaptoethanol (IMAC) and 5 mM DTT (SEC) in the respective buffers.

**Pre-crRNA cleavage by RcCas13a.** For the in vitro cleavage assay 0.625 µM or 0.208 µM 3′-FAM-labeled pre-crRNA (Ella Biotech; sequence in Supplementary Table S2) was incubated with 2 µM RcCas13a in cleavage buffer (20 mM Hepes pH

8.0, 100 mM NaCl, 50 mM KCl, 5% (v/v) glycerol, 0.8 µg/ml BSA) for 60–95 min at 37 °C in the absence or presence of 2 mM EDTA. Reactions were quenched by the addition of Urea loading buffer (New England Biolabs) and incubation at 75 °C for 15 min followed by analysis using denaturing polyacrylamide gel-electrophoresis (20% 19:1 acrylamid: bisacrylamid, 8% (w/v) urea, Tris-boric acid-EDTA, Carl Roth at 200 V). In gel fluorescence was analyzed with a GE Healthcare Amersham AI680 Imager.

**MALDI-TOF analysis of Cas13a pre-crRNA processing products**. Processed pre-crRNA samples for MALDI experiments were obtained by incubation of 3′-FAM-labeled pre-crRNA or respective desoxy-modified pre-crRNA (dA-modified/dC-modified) (24 µM; or 15 µM; Ella Biotech; sequences in Supplementary Table S2) with RcCas13a (33 µM) in cleavage buffer for 3 h at 37 °C. The Cas13a protein was removed by heat denaturation (95 °C; 10 min) followed by centrifugation (16,000 × g; 10 min; 4 °C. Unprocessed pre-crRNA (100 µM in $H_2O$) served as control. MALDI-TOF mass spectra were recorded on a Bruker Autoflex II. For the measurements, the samples were desalted against dd$H_2O$ using a 0.025 µm VSWP membrane filter (Merck Millipore) and then co-crystallized with a matrix solution (0.7 M 3-hydroxypicolinic acid, 0.07 M diammonium citrate in 1:1 $H_2O$/MeCN).

**In vivo activity assay**. The sequences encoding the luciferase subunits *luxA, luxB, luxC, luxD* and *luxE* from *Photorhabdus luminescens* (Uniprot IDs P23146, P19840, P23113, P23148, P19482) were cloned into pASK-IBA5 (IBA Lifesciences), under control of an anhydrotetracycline (aTc) inducible promoter. The gene encoding RcCas13a was cloned alone or together with CRISPR loci (or loci variant) targeting the *luxABCDE* genes (sequence in Supplementary Table S2) into pBAD RSF1031K, under control of the L-arabinose (ara) inducible pBAD promoter. In order to analyze bystander RNA degradation, the sequence encoding eGFP in fusion with a C-terminal ssrA degradation tag was cloned into pACYC, inducible with IPTG. Plasmids according to Supplementary Table S4 were transformed into *E. coli* JM109 (DE3) (NEB). Freshly prepared chemically defined EZ medium (Teknova), using 2% (v/v) glycerol as nutrient source (instead of glucose) and supplemented with the respective antibiotics, was inoculated from overnight cultures in triplicates to an $OD_{600}$ of 0.2. Cells were grown in a transparent 96-well plate (Greiner bio-one) with lid at 30 °C under agitation until an $OD_{600}$ of 0.55 was reached. RcCas13a expression was induced with a final concentration of 0.1% (w/v) L-arabinose. Cells were further grown until the $OD_{600}$ reached 0.7 and the expression of the reporter genes were induced with a final concentration of 0.2 mg/l aTc and 0.5 mM IPTG, respectively. Cell growth, the luminescence and the fluorescence were monitored for 24 h at 30 °C under agitation in a Spark 10 M plate reader (Tecan). All experiments were performed in three independent replicates.

**Protein crystallization and structure determination**. Crystals of the RcCas13a-crRNA binary complex were obtained by mixing 1 volume of protein (12 mg/ml) and 2 volumes of precipitant 100 mM 2-(N-morpholino)ethanesulfonic acid (MES) pH 6.5 and 25% (w/v) polyethylenglycol 6000 (PEG6000) at 20 °C. Crystals of the SeMet-labeled RcCas13a:crRNA binary complex were obtained by mixing 1 volume of protein (7.5 mg/ml) and 2 volumes of precipitant (100 mM MES pH 6.0, 200 mM NaCl and 20% (w/v) PEG6000) at 20 °C. Crystals were harvested using nylon loops (Hampton Research), soaked in artificial mother liquor supplemented with 30% (w/v) ethylene glycol as cryo-protectant and flash-cooled in liquid nitrogen. X-ray diffraction data were collected at beamlines PXI—X06SA of the Swiss Light Source in Villigen, Switzerland, and ID30A-1 (MASSIF-1) of the European Synchrotron Radiation Facility in Grenoble, France. Samples were cooled during data collection at 100 K with liquid nitrogen. Single anomalous dispersion (SAD) data on the SeMet-derivatised complex crystal were collected at 0.979 Å, native data were collected at 0.966 Å. Data collection, data processing and structure refinement statistics can be found in Table 1. Data were processed with XDS[34] and phasing was performed by SAD with HKL2MAP[35,36] and CRANK2[37]. Manual building of the model was done in COOT[38], refinement was carried out with PHENIX[39] and PDB_Redo[40], and structural figures were prepared with PyMol[41]. The final model was deposited at the Protein Data Bank with the accession code 7OS0.

**Analysis of Cas13a by dynamic light scattering (DLS)**. Purified Cas13a protein samples were analyzed by DLS. DLS measurements were performed using a DynaPro™ Nanostar™ DLS system (Wyatt Technology, Santa Barbara, CA) and analysis was done by using the DYNAMICS® software (version 8.0.0.77) provided by the manufacturer. Results from ten consecutive measurements at 25 °C (laser wavelength 661.3 nm) each with an acquisition time of 5 s and a read interval of 1 s were averaged for every protein sample. For the solution viscosity-parameter the viscosity of water at 20 °C (1.000 cP) was used.

**Statistics and reproducibility**. For experimental reproducibility of in vivo experiments, standard error of the mean was calculated using three independent biological experiments ($n = 3$). The error bars were calculated as standard deviation of three independent experiments using GraphPad Prism software (version 9.3.0 for Windows, San Diego, California USA, www.graphpad.com). For

visualization a scatter plot with bar (mean with SD) was chosen. Before statistical evaluation, fluorescence and luminescence values were normalized to the respective $OD_{600}$. The results were set in relation to each other as a percentage.

**Reporting summary**. Further information on research design is available in the Nature Research Reporting Summary linked to this article.

## Data availability
Structure coordinates and structure factor-files have been deposited in the PDB data bank (PDB code 7OS0). Source data for the graphs in Fig. 4b, c and data for the luminescence, fluorescence and bacterial growth measurements are available in Supplementary Data 1. Any remaining information can be obtained from the corresponding author upon reasonable request.

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

## Acknowledgements

This work was financially supported by the Deutsche Forschungsgemeinschaft (SCHN 1273-5/SPP 2141, SCHN 1273-6; Center for Integrated Protein Science Munich (CIPSM, EXC114) and SFB 1309). We thank the European Synchrotron Radiation Facility (ESRF) and the Swiss Light Source (SLS) for beamtime, and the staff of beamlines PX I (SLS), MASSIF-1 and ID23-2 (ESRF) for setting up the beamlines for data collection. We also thank Christopher Scheidler and Philipp Merz for assistance with protein purification for some activity assays.

## Author contributions

L.M.K. designed experiments, optimized protein purification and crystallization, determined the structure, performed in vivo activity assays. M.K.W. produced mutant proteins, carried out pre-crRNA processing and in vivo assays. L.S.R. carried out MALDI-ToF measurements, S.S. supervised the project, planed experiments, analyzed data, and wrote first draft of the manuscript. All authors discussed the results and edited the final manuscript version.

## Funding

## Competing interests

The authors declare no competing interests.
