## [Transparent Peer Review File · Communications Biology]

Reviewers' comments:

Reviewer #1 (Remarks to the Author):

In this manuscript Kick et al define the structure and mechanism of RNA cleavage by Cas13a from *Rhodobacter capsulatus*. Cas13 is a single-protein RNA-guided and RNA-activated CRISPR effector. Since their discovery a few years ago Cas13 enzymes have been adapted for a number of uses including RNA detection and targeted RNA decay. Prior to this work many structures of Cas13a bound to crRNA from a variety of other species have been solved and published. Moreover there are also several structures of Cas13b and Cas13d homologues. Collectively these structures established that Cas13 enzymes have two independent RNase sites (crRNA and HEPN) and that crRNA and target RNA binding triggers a conformational change that activates the HEPN nuclease active site for target RNA cleavage. The authors determined the structure of RcCas13a bound to a 58nt crRNA. Not surprisingly the overall structure of RcCas13a bound to crRNA is similar to other Cas13a enzymes, however there are notable differences in the Cas13a-crRNA interfaces. Aside from the HEPN domains Cas13 homologues are not well conserved so to determine the crRNA processing active site the authors mutated all the residues near the 5' end of the crRNA. Interestingly the authors identified three residues that are critical for crRNA processing: R1085, N1089, and H1113. The authors characterized the cleavage products by mass-spec and found that the crRNA active site produces cleavage products with a 2'3'-cyclic phosphate and 5'OH supporting an RNase A like transesterification mechanism. While the authors do not comment on this in the manuscript I think it is intriguing that both the crRNA and HEPN active sites use a similar reaction mechanism for RNA cleavage. Finally the authors investigated RcCas13a's ability to cleave RNA in bacteria and they found that the CTD which is not visible in their crystal structure is not important for cleavage. Overall, this manuscript is well written and provides new insight into crRNA processing by RcCas13a however I do have several concerns that need to be addressed.

1. My major concern with the paper is the experiments addressing the in vivo activity of Cas13a. These experiments were carried out with large domain deletions of Cas13. Removal of the disordered CTD is unlikely to cause any disruptions in the overall structure of Cas13 or RNA binding, however removal of the HEPN domains likely results in a protein that is not properly folded and unable to bind RNA. The authors should repeat these assays with specific HEPN mutants (R464, H649, R1052, H1057, R1085, H1113) rather than domain deletions.
2. The mass-spec data shows that RcCas13 cuts the 5' end of the crRNA at two different positions but the RNA gels shown in Fig. 3A do not have sufficient resolution to resolve these two products. Given the extensive interface b/w the crRNA and Cas13 it is unclear to this reviewer why RcCas13 would have more than one cleavage site. Is one of the cleavage sites preferred? Is this observed with any other Cas13 enzymes? Is this dependent upon the sequence of the 5' end of the crRNA? This finding of two cuts sites needs to be further explored.
3. Figure S6 provides a comparison of rcCas13 with other Cas13 structures, however it is impossible to identify differences in the structures as there are no labels and the crRNA and HEPN active sites are not identified/boxed. The discovery that RcCas13a has a completely distinct crRNA active site is one of the major findings of this paper and needs to be better highlighted in this figure.
4. The crRNA active site is within the second HEPN domain. HEPN nucleases are found across all walks of life and facilitate a number of different RNA processing events through a very small conserved RxxxxH motif. HEPN domains are composed of a core of 4 alpha helices but many have specialized features/extensions that support their diverse biological roles. Beyond just comparing RcCas13 to other Cas13a enzymes the authors should compare the RcCas13 HEPN domains to other HEPN family members. I'm not aware of any other HEPN domain that contain a second RNase active site so this is an interesting finding that warrants discussing in the paper.
5. In Fig. S10 the labels in C should be colored as in B to make it easier to distinguish.

Reviewer #3 (Remarks to the Author):

Cas13 proteins are Type VI CRISPR-Cas endonucleases that possess two distinct RNase activities: the pre-crRNA processing activity and the promiscuously trans-RNA cleavage activity. In the current study, Kick et al., determined the structure of *Rhodobacter capsulatus* (Rc) Cas13a-crRNA complex using X-ray crystallography. The high-resolution structural information and mutagenesis experiments allow them to infer the pre-crRNA cleavage mechanism of RcCas13a. Interestingly, the authors analyzed the pre-crRNA processing products of RcCas13a by MALDI-TOF mass spectrometry and identified 2'3'-cyclic phosphate as the Cas13a pre-crRNA processing intermediate for the first time, providing valuable insights into the mechanisms of pre-crRNA processing. Also, the authors evaluated the cis- and trans- cleavage activity of RcCas13a in *E. Coli* supporting previous observations that Cas13a activation induces growth arrest of host cells to prevent the rise of CRISPR-resistant phages. Although the discovery of 2'3'-cyclic phosphate is novel, some of the major conclusions have not been fully justified. Please address the following issues in the manuscript of next version:

Major issues:

The structural basis of crRNA maturation by LbaCas13a has been discussed in depth (Figure 4, Knott et al., 2017, NSMB), where a water molecule (observed in the crystal structure) is proposed to deprotonate the 2' hydroxyl of the RNA pentose ring in an acid-base catalysis reaction. In the current manuscript, the authors proposed two possible acid-base catalytic mechanisms: (I) H1113 act as a base and deprotonate the 2' hydroxyl group directly, and (II) "His 1113 abstracts a proton from a water molecule and that the resulting hydroxide ion acts as the effective base". However, the reason (Line 211- Line 214) to favor mechanism (I) is not convincing to me, because processing the pre-crRNA into mature crRNA is an intrinsic property of all Cas13 effectors. The following are two questions specific to the pre-crRNA cleavage activity of RcCas13a:

1. Since the RcCas13a-crRNA binary complex doesn't represent a pre-crRNA bound state, I suggest that the authors prepare a pre-crRNA cleavage deficient mutant of RcCas13a, and use this protein for crystallization and structural characterization. A similar strategy has been used for structural investigation of LbaCas13a (Knott et al., 2017, NSMB).

2. How similar are the catalytic pockets of RcCas13a and LbaCas13a? The authors should prepare a figure to show the pre-crRNA cleavage pocket of both RcCas13a and LbaCas13a, and discuss their different catalytic mechanisms.

3. Line 125- Line 127: "Nucleobases at positions 10-14 in the crRNA spacer region are solvent exposed and are presumably acting as the seed region for target RNA binding (Fig 1B), as shown for the homolog LbuCas13a". I would like to see experimental demonstration of the "seed region" of RcCas13a, rather than a speculation based on the homologous structure of LbuCas13a.

4. Supporting Figure S5. There is a poorly modeled region in the spacer RNA. What is the sequence of this region? Can the increased structural flexibility be a consequence of its solvent exposure? What is the relationship between the poorly modeled region and the seed region?

5. Line 193 - Line 195: The text is discussing about LbuCas13a, but reference #8 refers to a study about LbaCas13a. Please use the correct reference.

6. Figure 4 and Supporting Figure S14. The authors use "off-target cleavage" to describe the promiscuous trans-RNA cleavage activity which can be misleading to readers outside of the field. Please use a more accurate term instead (e.g. bystander cleavage, collateral cleavage, trans-ssRNA cleavage). Similarly, the sentence containing "off-target RNA degradation" in Line 358 should be adjusted accordingly.

7. Figure 4. Activation of Cas13a effector complex in E Coli. unleashes collateral cleavage activity and degrade ssRNAs in an indiscriminate manner. Because of the collateral effect, both GFP and Luc signal should be expected to diminish upon activation of Cas13a, and abolish of the HEPN domain can restore the signal, as is shown in Figure4-B. Therefore, the cartoon for cis-cleavage (GFP+, Luc-readout) does not reflect what actually happens in host cells. In addition to the luminescent and fluorescent assays, the authors should perform northern blot analysis to evaluate RNA degradation.

8. Figure 4. Cas13a can effectively knockdown target RNAs in a sequence specific manner, and the collateral effect is inhibited in mammalian cells by an unknown mechanism (Abudayeh et al., Nature, 2017). Therefore, the authors should consider using a mammalian system for in vivo cis- and trans-cleavage assays rather than using bacteria.

Minor issues:

9. Line 197: "an Arg and His located in the NTD are responsible for pre-crRNA cleavage." Which Arg and His are involved? Please specify their residue numbers.

10. Supporting Figure S6. Please give the PDB codes of all Cas13a structures used for 3D alignment in the figure legend.

First, we would like to thank the reviewers for their thorough reading of our manuscript, the valuable comments and constructive suggestions, which have allowed us to significantly improve the quality of the manuscript. Please find below our point-to-point-response

Reviewers' comments:

Reviewer #1 (Remarks to the Author):

In this manuscript Kick et al define the structure and mechanism of RNA cleavage by Cas13a from *Rhodobacter capsulatus*. Cas13 is a single-protein RNA-guided and RNA-activated CRISPR effector. Since their discovery a few years ago Cas13 enzymes have been adapted for a number of uses including RNA detection and targeted RNA decay. Prior to this work many structures of Cas13a bound to crRNA from a variety of other species have been solved and published. Moreover there are also several structures of Cas13b and Cas13d homologues. Collectively these structures established that Cas13 enzymes have two independent RNase sites (crRNA and HEPN) and that crRNA and target RNA binding triggers a conformational change that activates the HEPN nuclease active site for target RNA cleavage. The authors determined the structure of RcCas13a bound to a 58nt crRNA. Not surprisingly the overall structure of RcCas13a bound to crRNA is similar to other Cas13a enzymes, however there are notable differences in the Cas13a-crRNA interfaces. Aside from the HEPN domains Cas13 homologues are not well conserved so to determine the crRNA processing active site the authors mutated all the residues near the 5' end of the crRNA. Interestingly the authors identified three residues that are critical for crRNA processing: R1085, N1089, and H1113. The authors characterized the cleavage products by mass-spec and found that the crRNA active site produces cleavage products with a 2'3'-cyclic phosphate and 5'OH supporting an RNase A like transesterification mechanism. While the authors do not comment on this in the manuscript I think it is intriguing that both the crRNA and HEPN active sites use a similar reaction mechanism for RNA cleavage. Finally, the authors investigated RcCas13a's ability to cleave RNA in bacteria and they found that the CTD which is not visible in their crystal structure is not important for cleavage. Overall, this manuscript is well written and provides new insight into crRNA processing by RcCas13a however I do have several concerns that need to be addressed.

1. My major concern with the paper is the experiments addressing the in vivo activity of Cas13a. These experiments were carried out with large domain deletions of Cas13. Removal of the disordered CTD is unlikely to cause any disruptions in the overall structure of Cas13 or RNA binding, however removal of the HEPN domains likely results in a protein that is not properly folded and unable to bind RNA. The authors should repeat these assays with specific HEPN mutants (R464, H649, R1052, H1057, R1085, H1113) rather than domain deletions.

We are sorry that it was not clear from the text that " Δ HEPN1, Δ HEPN2, and Δ HEPN1+2 correspond to the specific point mutations of the catalytic residues and not whole domain deletions. Thus Δ HEPN1 = RcCas13A R464A/H469A; Δ HEPN2 = R1052A/H1057A; Δ HEPN1+2 = RcCas13A R464A/H469A/R1052A/H1057A. Since this would be rather long label to use in the Figure legends we have altered that in Fig. 4 and legend to: H1* = RcCas13A R464A/H469A, H2* = R1052A/H1057A. H1+2* = RcCas13A R464A/H469A, R1052A/H1057A. We hope this is clear now from the labelling and text.

2. The mass-spec data shows that RcCas13 cuts the 5' end of the crRNA at two different positions but the RNA gels shown in Fig. 3A do not have sufficient resolution to resolve these two products. Given the extensive interface b/w the crRNA and Cas13 it is unclear to this reviewer why RcCas13 would have more than one cleavage site. Is one of the cleavage sites preferred? Is this observed with

any other Cas13 enzymes? Is this dependent upon the sequence of the 5' end of the crRNA? This finding of two cuts sites needs to be further explored.

We agree with the reviewer that the exact cleavage position can not be deduced from the RNA gel analysis due to its limited resolution, particular if minor products are below the detection limit of fluorescent labelling. Therefore we turned to analysis by mass spectrometry (MALDI-ToF), which provides very high sensitivity and precision and is commonly used for the analysis of oligonucleotides (doi: 10.1016/j.ab.2008.04.031) In order to address the cleavage site preference further as well as to obtain further mechanistic insight into the mechanism, we used synthetic RNA oligonucleotides where the RNA nucleobase at the cleavage position was substituted by its DNA counterpart (dA or dC, respectively), which should make the attack of the 2'OH only at one of the two cleavage position possible. Indeed, we observe that with the strand either containing the dA or dC, respectively, cleavage can only be observed at one position. (Supporting Fig 13F+G) To the best of our knowledge crRNA processing at different sites has not been reported for other Cas endonucleases. It should be noted that this is only visible by using high resolution methods such as MALDI-ToF. So far only RNA-gel electrophoresis (in combination with ³²P labelled substrate RNA has been used.

3. Figure S6 provides a comparison of rcCas13 with other Cas13 structures, however it is impossible to identify differences in the structures as there are no labels and the crRNA and HEPN active sites are not identified/boxed. The discovery that RcCas13a has a completely distinct crRNA active site is one of the major findings of this paper and needs to be better highlighted in this figure. 4. The crRNA active site is within the second HEPN domain. HEPN nucleases are found across all walks of life and facilitate a number of different RNA processing events through a very small conserved RxxxxH motif. HEPN domains are composed of a core of 4 alpha helices but many have specialized features/extensions that support their diverse biological roles. Beyond just comparing RcCas13 to other Cas13a enzymes the authors should compare the RcCas13 HEPN domains to other HEPN family members. I'm not aware of any other HEPN domain that contain a second RNase active site so this is an interesting finding that warrants discussing in the paper.

We thank the reviewer for pointing out this intriguing observation. We have now added an addition figure to the SI (Supporting Fig. S8) where we compare the HEPN domain of RcCas13a to UrCas13d and other HEPN-domain containing proteins. Moreover, we have added a figure to compare the crRNA-processing site of RcCas13a and LbaCas13a. In addition, we have added the following discussion to the manuscript:

Introduction:

“Albeit HEPN-domains do not share a significant sequence conservation, they are characterized by an α -helical-folding topology and a conserved Arg and His sequence motif (R(x)4-6H, where x can be any residue). Two HEPN-domains have to align the Arg and His residues, either through dimerization of the two domains contained within one protein, like in Cas13 endonucleases, or of two HEPN-domain proteins, two form the catalytic tetrad required for metal-independent endonuclease activity.^{21,22}”

Discussion:

“While in the homolog from *L. shahii* (LshCas13a) Arg 438 and Lys 441 located in the NTD are responsible for pre-crRNA cleavage,⁹ Dounda co-worker report for LbaCas13a that Lys 1305 and Asn 1232 located in the HEPN2 domain are required/impact for pre-crRNA processing.¹⁰ While in the homolog from *L. shahii* (LshCas13a) Arg 438 and Lys 441 located in the NTD are responsible for pre-crRNA cleavage,⁹ Dounda co-worker report for LbaCas13a that Lys 1305 and Asn 1232 located in the HEPN2 domain are required/impact for pre-crRNA processing.¹⁰ In addition, in the LbaCas13a complex the 5' flank of the pre-crRNA and crRNA are tightly held by the surrounding amino acid side chains. In contrast, the equivalent site in the RcCas13a complex is

more open, allowing more flexibility for the 5' flank of the crRNA (Supporting Fig. S14). Nevertheless, LbaCas13a and RcaCas13a both utilize residues located in the HEPN2 domain for pre-crRNA processing. Cas13d from uncultured *Ruminococcus* sp. (UrCas13d), another member of the Type VI family of Cas endonucleases, also requires two residues, Arg 802 and Lys 905 located in its HEPN2 domain for pre-crRNA processing.⁵ These are to our knowledge to date the only examples of catalytic activity HEPN domains that contain a second RNase active site (Supporting Fig. S14)”

5. In Fig. S10 the labels in C should be colored as in B to make it easier to distinguish. The labels were colored accordingly

Reviewer #3 (Remarks to the Author):

Cas13 proteins are Type VI CRISPR-Cas endonucleases that possess two distinct RNase activities: the pre-crRNA processing activity and the promiscuously trans-RNA cleavage activity. In the current study, Kick et al., determined the structure of *Rhodobacter capsulatus* (Rc) Cas13a-crRNA complex using X-ray crystallography. The high-resolution structural information and mutagenesis experiments allow them to infer the pre-crRNA cleavage mechanism of RcCas13a. Interestingly, the authors analyzed the pre-crRNA processing products of RcCas13a by MALDI-TOF mass spectrometry and identified 2'3'-cyclic phosphate as the Cas13a pre-crRNA processing intermediate for the first time, providing valuable insights into the mechanisms of pre-crRNA processing. Also, the authors evaluated the cis- and trans- cleavage activity of RcCas13a in *E. Coli* supporting previous observations that Cas13a activation induces growth arrest of host cells to prevent the rise of CRISP-resistant phages.

Although the discovery of 2'3'-cyclic phosphate is novel, some of the major conclusions have not been fully justified. Please address the following issues in the manuscript of next version:

Major issues:

The structural basis of crRNA maturation by LbaCas13a has been discussed in depth (Figure 4, Knott et al., 2017, NSMB), where a water molecule (observed in the crystal structure) is proposed to deprotonate the 2' hydroxyl of the RNA pentose ring in an acid-base catalysis reaction. In the current manuscript, the authors proposed two possible acid-base catalytic mechanisms: (I) H1113 act as a base and deprotonate the 2' hydroxyl group directly, and (II) ‘His 1113 abstracts a proton from a water molecule and that the resulting hydroxide ion acts as the effective base’. However, the reason (Line 211- Line 214) to favor mechanism (I) is not convincing to me, because processing the pre-crRNA into mature crRNA is an intrinsic property of all Cas13 effectors. The following are two questions specific to the pre-crRNA cleavage activity of RcCas13a:

1. Since the RcCas13a-crRNA binary complex doesn't represent a pre-crRNA bound state, I suggest that the authors prepare a pre-crRNA cleavage deficient mutant of RcCas13a, and use this protein for crystallization and structural characterization. A similar strategy has been used for structural investigation of LbaCas13a (Knott et al., 2017, NSMB).

We agree with the reviewer that a crystal structure of pre-crRNA bound state would provide a very valuable snap-shot of the amino acid side chains in the direct vicinity of the processing site. Unfortunately, this is not possible since successful crystallisation of the RcCas13-crRNA complex requires co-expression of the crRNA with RcCas13a and co-purification, which includes processing of the crRNA by RcCas13a upon co-expression within *E. coli*. Initially we attempted to obtain the RcCas13a-crRNA complex structure by incubating RcCas13a either with crRNA from *in vitro* transcription or synthetic crRNA, followed by size exclusion chromatography, but could not obtain crystals.

However, we have carried out additional experiments, where we mutated N1089 to Asp and analysed the cleavage product, as well as the cleavage products of the N1089A mutant by MALDI-TOF. Here we observe that while mutation to Asp does not alter the cleavage position (MALDI-ToF analysis) or impacts on cleavage efficiency (as analysed by gel electrophoresis), that the mutation to Ala alters the cleavage position. (Supporting Fig. S15) This indicates that Asn1089 is involved in stabilising the pre-crRNA for cleavage.

Moreover, using synthetic pre-crRNA where we substituted the ribose by desoxyribose at the cleavage positions (A→dA and C→ dC, respectively) we could show that the 2'OH is responsible for the nucleophilic attack onto the scissile phosphate (Supporting Fig. S15)

2. How similar are the catalytic pockets of RcCas13a and LbaCas13a? The authors should prepare a figure to show the pre-crRNA cleavage pocket of both RcCas13a and LbaCas13a, and discuss their different catalytic mechanisms.

We have added an additional figure to the Supporting Information (Supporting Fig. X) for comparing of the pre-crRNA processing pocket of the complexes of RcCas13a-crRNA, LbaCas13a-crRNA and the LbaCas13a-His328A-pr-crRNA complexes. This shows that the binding site for the 5' flank in RcCas13a is much more open than in LbCas13a. We have also reworded the section where we discuss the cleavage mechanism and also discuss the differences between RcCas13a and LbaCas13a:

“To gain more insights into pre-crRNA processing by RcCas13, we analyzed the reaction products of the in vitro pre-crRNA processing reaction by MALDI-TOF (Matrix Assisted Laser Desorption Ionisation- Time Of Flight) mass spectrometry, which possesses much higher sensitivity and precision in comparison to gel-electrophoresis. This revealed that RcCas13a cleaves either 4 or 5 nt 5' to the stem loop, generating a 2',3'-cyclic 3' phosphate (2',3' cP, Fig. 3E, Supporting Fig. 14). Given that the reaction product is a 2',3'-cP, the 2'-oxygen next to the scissile phosphate has to carry out the nucleophilic attack and is thus transferred to the 5' phosphate. Substitution of the ribose by the 2' deoxyribose (dA or dC) in synthetic pre-crRNA strands abrogates cleavage at this positions (Supporting Fig. 13), providing further support for the importance of the nucleophilic attack by the 2'OH. Prerequisite for this transfer reaction is deprotonation of the 2'OH group, possibly by the essential His 1113 acting as the base. However, it is also possible that His 1113 abstracts a proton from a water molecule and that the resulting hydroxide ion acts as the effective base, since our structure shows RcCas13a in complex with the processed, mature crRNA and not the pre-crRNA bound state. Asn 1089 could play a role in the stabilization of the penta-covalent reaction intermediate and/or positioning of the 5' flank, since mutation of Asn 1089 to Asp does not impact on the cleavage position but mutation to Ala leads to aberrant cleavage 6 nt's 5' to the stem loop (Supporting Fig. 14). Arg 1085 could act as the acid and protonates the 5'-oxygen leaving group (Fig. 3F). Due to the high intrinsic pKa value of the Arginine's guanidinium group (pKa 13.8 ± 0.1) and delocalization of the positive charge over many atoms involved in a conjugated π-system, the arginine side chain has the unusual ability to retain its charge under all physiological conditions and can thus function as an acid catalyst.^{29,30} As already observed by gel electrophoresis, the RcCas13aR1085A mutant does not show any pre-crRNA cleavage activity in the MALDI-analysis of the reaction products (Supporting Fig. 14). As stated above, in our mutational analysis of residues lining the pre-crRNA processing site in RcCas13a, we could identify Arg 1085 and His 1113 as being clearly essential, and Asn 1089 modulating crRNA cleavage. In contrast, for LbaCas13a, which possesses a narrower pre-crRNA binding site (Supporting Fig. X), it was shown that Trp 325 and Asn 1232 are the likely candidates to set up the 2'OH of the ribose of the scissile base in the pre-crRNA for nucleophilic attack. Than an activated water molecule, together with Lys 1305, His 328, Lys432, and Lys 1320 participate in the proton transfer reactions.¹⁰”

3. Line 125- Line 127: “Nucleobases at positions 10-14 in the

crRNA spacer region are solvent exposed and are presumably acting as the seed region for target RNA binding (Fig 1B), as shown for the homolog LbuCas13a". I would like to see experimental demonstration of the "seed region" of RcCas13a, rather than a speculation based on the homologous structure of LbuCas13a.

We have now performed additional activity assays in *E.coli* where we could show that scrambling of the proposed seed region in the spacer sequence nt 10-14 of the *luxABCDE*-targeting crRNA abolished RcCas13a-mediated reduction of luciferase luminescence (Fig. 4B).

4. Supporting Figure S5. There is a poorly modeled region in the spacer RNA. What is the sequence of this region? Can the increased structural flexibility be a consequence of its solvent exposure? What is the relationship between the poorly modeled region and the seed region?

Yes, we could not confidentially place the model in the electron density in this region do to the intrinsic flexibility, most likely caused by solvent exposure. We have added to the Figure legend of Supporting Fig. S5: "Parts of the spacer region are flexible and not well defined in the electron density most likely do to solvent exposure." The full sequence of the crRNA is shown in Supporting Table S3.

5. Line 193 - Line 195: The text is discussing about LbuCas13a, but reference #8 refers to a study about LbaCas13a. Please use the correct reference.

Corrected.

6. Figure 4 and Supporting Figure S14. The authors use "off-target cleavage" to describe the promiscuous trans-RNA cleavage activity which can be misleading to readers outside of the field. Please use a more accurate term instead (e.g. bystander cleavage, collateral cleavage, trans-ssRNA cleavage). Similarly, the sentence containing "off-target RNA degradation" in Line 358 should be adjusted accordingly.

We use now the term bystander cleavage in Fig 4., Supporting Fig S14 and the methods section.

7. Figure 4. Activation of Cas13a effector complex in E Coli. unleashes collateral cleavage activity and degrade ssRNAs in an indiscriminate manner. Because of the collateral effect, both GFP and Luc signal should be expected to diminish upon activation of Cas13a, and abolish of the HEPN domain can restore the signal, as is shown in Figure4-B. Therefore, the cartoon for cis-cleavage (GFP+, Luc- readout) does not reflect what actually happens in host cells. In addition to the luminescent and fluorescent assays, the authors should perform northern blot analysis to evaluate RNA degradation.

We thank the reviewer to pointing this out and have adapted Fig. 4 and Supporting Fig SX accordingly. Unfortunately, Northernblot analysis would not allow us to discriminate between target and bystander cleavage since upon activation of Cas13a all ssRNA within the cell will get degraded. Thus we would not have a reference point ("housekeeping RNA") for quantification required by Northernblot analysis (or qRT-PCR). Thus analysis of the GFP fluorescence is representative for bystander ssRNA cleavage. However, as stated above we have now carried out an additional experiment, were the sequence of the seed region (nt 10-14) in the crRNA was scrambled. Here we do not observe any changes in luciferase expression, thus clearly showing that reduction of luciferase expression is dependent on the specific crRNA sequence (addition to Fig 4B).

8. Figure 4. Cas13a can effectively knockdown target RNAs in a sequence specific manner, and the collateral effect is inhibited in mammalian cells by an unknown mechanism (Abudayeh et al.,

Nature, 2017). Therefore, the authors should consider using a mammalian system for in vivo cis- and trans- cleavage assays rather than using bacteria.

Albeit it would be interesting to see whether RcCas13 could be used for modulation of gene expression in mammalian cells as observed for other members of the type IV-family of Cas13 endonucleases, this is beyond the scope of the present manuscript. The aim of our study is to shed light on the structure-function relationship and mechanism of RcCas13a.

Minor issues:

9. Line 197: “an Arg and His located in the NTD are responsible for pre-crRNA cleavage.” Which Arg and His are involved? Please specify their residue numbers.

Residue numbers have been added accordingly.

10. Supporting Figure S6. Please give the PDB codes of all Cas13a structures used for 3D alignment in the figure legend.

All PDB codes have been added to the figure legends

REVIEWERS' COMMENTS:

Reviewer #1 (Remarks to the Author):

The reviewers have adequately addressed all my concerns and I enthusiastically support publication of this manuscript in Communications Biology. In particular, I appreciate the additional studies looking at the Rccas13 cleavage sites and the new supplemental figures comparing Rccas13 to other Cas13 nucleases and other HEPN nucleases.

Reviewer #3 (Remarks to the Author):

My questions have been addressed. Current manuscript is good for publication.

REVIEWERS' COMMENTS:

Reviewer #1 (Remarks to the Author):

The reviewers have adequately addressed all my concerns and I enthusiastically support publication of this manuscript in Communications Biology. In particular, I appreciate the additional studies looking at the RcCas13 cleavage sites and the new supplemental figures comparing RcCas13 to other Cas13 nucleases and other HEPN nucleases.

Reviewer #3 (Remarks to the Author):

My questions have been addressed. Current manuscript is good for publication.

We are again grateful for the comments and suggestions made by the reviewers. The comments and constructive suggestions have allowed us to significantly improve the quality of the manuscript